# Broad geographical circulation of a novel vesiculovirus in bats in the Mediterranean region

Dong-Sheng Luo[1,2]*, Markéta Harazim[3], Corinne Maufrais[4], Simon Bonas[1], Natalia Martinkova[3], Aude Lalis[5], Emmanuel Nakouné[6], Edgard Valéry Adjogoua[7], Mory Douno[8], Blaise Kadjo[9], Marc López-Roig[10,11], Jiri Pikula[12], Zheng-Li Shi[13], Hervé Bourhy[1], Jordi Serra-Cobo[10,11], Laurent Dacheux [1]¤*

1 Institut Pasteur, Université Paris Cité, Unit Lyssavirus Epidemiology and Neuropathology, Paris, France, 2 Department of Infection Biology, London School of Hygiene & Tropical Medicine, London, United Kingdom, 3 Institute of Vertebrate Biology, The Czech Academy of Sciences, Brno, Czechia, 4 Institut Pasteur, Université Paris Cité, Bioinformatics and Biostatistics Hub, Paris, France, 5 Institut de Systématique, Evolution, Biodiversité (ISYEB), Muséum National d'Histoire Naturelle, CNRS, SU, EPHE-PSL, UA, Paris, France, 6 Virology Department, Institut Pasteur de Bangui, Bangui, Central African Republic, 7 Department of Epidemic Viruses, Institut Pasteur de Côte d'Ivoire, Abidjan, Côte d'Ivoire, 8 Centre de Gestion de l'Environnement des Monts Nimba et Simandou (CEGENS) Lola/ Ministère de l'Environnement et du Développement Durable, Conakry, Guinée-Conakry, 9 Université Félix-Houphouët-Boigny, UFR Biosciences, Abidjan, Côte d'Ivoire, 10 Departament de Biologia Evolutiva, Ecologia i Ciències Ambientals, Facultat de Biologia, Barcelona, Spain, 11 Institut de Reserca de Biodiversitat (IRBio), Universitat de Barcelona, Barcelona, Spain, 12 Department of Ecology and Diseases of Zoo Animals, Game, Fish and Bees, University of Veterinary Sciences Brno, Brno, Czechia, 13 Guangzhou National Laboratory, Guangzhou International Bio Island, Guangzhou, China

¤Current address: Institut Pasteur, Université Paris Cité, Unit Environment and Infectious Risks, Paris, France

* dongshengluo@outlook.com (DSL); laurent.dacheux@pasteur.fr (LD)

## Abstract

Bats are the natural reservoirs for a variety of emerging and re-emerging viruses. Among them, rabies virus (genus *Lyssavirus*, family *Rhabdoviridae*) is one of the first and most emblematic described in these animals. Since its first description, several new bat lyssaviruses have been regularly identified. In addition to lyssaviruses, other bat rhabdoviruses have also been discovered, including members of the genera *Vesiculovirus*, *Ledantevirus* and, more recently, *Alphanemrhavirus* and *Tupavirus*. However, the family *Rhabdoviridae* is one of the most abundant and diverse viral families, with 434 officially recognized species, divided into 5 subfamilies and 56 different genera. The number of rhabdoviruses associated with bats is therefore probably higher than that currently available. In this study, we first developed and validated a combined nested RT-qPCR technique (pan-rhabdo RT-nqPCR) dedicated to the broad detection of animal rhabdoviruses. After validation, this technique was used for a large retrospective screening of archival bat samples (n = 1962), including blood (n = 816), brain (n = 723) and oral swab (n = 423). These samples were collected from various bat species over a 12-year period (2007–2019) in 9 different countries in Europe and Africa. A total of 23 samples (1.2%) from bat species *Miniopterus schreibersii*, *Rhinolophus euryale* and *Rhinolophus ferrumequinum* tested positive for rhabdovirus infection, including 17 (2.1%) blood and 6 (1.4%)

**Data availability statement:** Sequences of the Mediterranean bat virus are publicly available in GenBank (accession numbers: MW557328-MW557343). All other relevant data are in the manuscript and its Supporting information files.

**Funding:** This work was jointly funded by Campus France (French Ministry of Higher Education and Research) and China Scholarship Council through the PHC Cai Yuanpei 2016 program under grant number 36724VF, as well as by Institut Pasteur, Paris to DSL. The funders had no role in study design, data collection and analysis, decision to publish, or preparation of the manuscript.

**Competing interests:** The authors have declared that no competing interests exist.

oral swab samples, all collected from bats originating from the Mediterranean region. Complete virus genome sequences were obtained by next-generation sequencing for most of the positive samples. Molecular and phylogenetic analysis of these sequences demonstrated that the virus isolates, named Mediterranean bat virus (MBV), were closely related and represented a new species, *Mediterranean vesiculovirus*, within the *genus Vesiculovirus*. MBV was more specifically related to other bat vesiculoviruses previously described from China and North America, together clustering into a distinct group of bat viruses within this genus. Interestingly, our results suggest that MBV is widespread, at least in the western part of the Mediterranean region, where it circulates in the blood of several bat species. These results expand the host range and viral diversity of bat vesiculoviruses, and pave the way for further studies to determine the transmission route and dissemination dynamics of these viruses in bat colonies, as well as to assess their potential threat to public health.

## Author summary

Bats are the natural hosts for a wide variety of viruses, particularly those belonging to the family *Rhabdoviridae*, which include the lyssaviruses, the etiological agents of rabies. This viral family is one of the most abundant and diverse, with 434 officially recognized species. However, the current diversity of these viruses in bats remains poorly understood.

To address this, we developed a new method for detecting rhabdoviruses in bat samples collected over a period of 12 years in several European and African countries. From nearly 2,000 samples, we discovered a new rhabdovirus, which was named Mediterranean bat virus (MBV).

MBV has been detected in the blood and oral swabs of specific bat species (*Rhinolophus* sp. and *Miniopterus schreibersii*) living on both sides of the Mediterranean region. Based on phylogenetic analysis, we determined that MBV represented a unique group within the genus *Vesiculovirus*, but still related to other bat vesiculoviruses discovered in China and North America.

These results expand the host range and viral diversity of bat rhabdoviruses, and pave the way for further studies to determine the transmission route and dissemination dynamics of these viruses in bat colonies, as well as to assess their potential threat to public health.

## 1. Introduction

Bats belong to the order Chiroptera, which represents the second largest mammalian order, with 1474 species extant to date, belonging to 21 bat families (https://www.mammaldiversity.org/). These animals have many specific biological and ecological

characteristics, including various unique living habits and extensive geographical distribution [1–3]. For example, the *Miniopterus schreibersii* bat species has an extensive geographical distribution, occurring in the Middle East, in Central to Southeast Asia, in North Africa and in southern Europe, with the largest populations found in the warmer Mediterranean area.

In addition, bats are attracting growing interest from the scientific community, particularly in terms of public health. Indeed, in the past few decades, hundreds of virus species belonging to various families have been detected or isolated in bats, suggesting that these animals represent a major animal reservoir [4–7]. Among them, numerous emerging or re-emerging infectious diseases were demonstrated or suspected to be bat-related, including coronavirus-based diseases with severe acute respiratory syndrome (SARS) [8,9], Middle-East respiratory syndrome (MERS) [10] and more recently the 2019 coronavirus pandemic (COVID-19) [11], as well as fatal hemorrhagic diseases caused by filoviruses (Ebola and Marburg viruses) [12,13] and henipaviruses (Hendra and Nipah viruses) [14–16]. Furthermore, as the research on bat-associated viruses progresses, this viral diversity continues to expand, particularly in other families of viruses [6]. However, some of these virus families still remain poorly investigated, such as the family *Rhabdoviridae* outside of lyssaviruses.

The family *Rhabdoviridae* belongs to the order *Mononegavirales* and encompasses 434 different species distributed among 4 subfamilies and 56 genera (https://ictv.global/taxonomy) [17,18]. Rhabdoviruses are mostly characterized by a linear, negative-sense, single-stranded RNA genome of around 10–16 kb with five canonical genes encoding the nucleoprotein (N), the phosphoprotein (P), the matrix protein (M), the glycoprotein (G), and the RNA-dependent RNA polymerase (L) [17]. These viruses exhibit a large ecological diversity, infecting plants, insects or various vertebrates such as mammals [17,19–21].

Among mammals, bats have so far been associated with four different rhabdovirus genera. The first and most emblematic is the genus *Lyssavirus*, which includes rabies virus (RABV) [22,23]. RABV was the first bat virus described, and nearly all the different lyssavirus species or tentative species have been isolated from bats [24], suggesting that these animals are their original and natural reservoir [25]. The two other main genera are *Vesiculovirus* and *Ledantevirus*, with bat viruses reported in North America and China [26–28] or in several countries of Africa, Asia, Europe or North America [4,29–35], respectively. Interestingly, many ledanteviruses have also been isolated from arthropods feeding on bats, suggesting that they can act as arboviruses [33,36]. Recently, one novel rhabdovirus, Sodak rhabdovirus 2 (SDRV2), was identified in North American bats, and was classified as a member of the *Alphanemrhavirus* genus [37]. In addition, an individual virome analysis of Chinese bats identified putative new virus members among the *Tupavirus* genus [38].

All together, these data suggest that bats are playing an important role in the diffusion and persistence of rhabdovirus, with potential risks of human exposure. Apart from bat lyssaviruses, recent examples of potential zoonotic bat rhabdoviruses have been described. For instance, members of genus *Ledantevirus* identified in bats in China (with Rhinolophus rhabdovirus DPuer in *Rhinolophus affinis*) and Ghana (with Kumasi rhabdovirus in *Eidolon helvum*) have shown the ability to infect human cells *in vitro* [31,35]. In addition, populations living near these bats showed specific antibodies, demonstrating natural exposure to these viruses. The risk of transmission of bat rhabdovirus was also underlined with the identification of American bat vesiculovirus (ABV) in bats which were in contact with humans in the USA [39].

Rhabdovirus evolution is characterized by frequent host changes and genetic diversification, due to the simplicity of their genomic structure and their high mutation rate [40]. The discovery of new rhabdoviruses in wild animals, particularly in bats, raises concerns about their potential for transmission to humans and domestic animals. Their wide host range, ecological plasticity and ability to adapt to new environments highlight the importance of surveillance to identify any potential zoonotic emergence in the future.

Most of these studies have been carried out through the implementation of complex techniques that are often costly and time-consuming, such as metagenomic approaches, or following viral isolation experiments. The development and validation of a molecular tool enabling pan-rhabdovirus detection in a fast and reliable way would be a considerable asset to more easily explore the diversity of rhabdoviruses in these animals, then assess their zoonotic risk. Due to the high

genetic diversity within the family *Rhabdoviridae*, the development of such generic molecular tools with a satisfactory balance between specificity and sensitivity, remains a key challenge. Some innovative approaches have been carried out [41], but most of existing tools rely on PCR techniques, dedicated to a specific species or genus such as lyssaviruses [28,42], or with an extended spectrum of detection [43–45]. Despite these efforts, the spectrum of detection offered by these techniques remains limited and needs further improvement.

To achieve this goal, our study aimed to develop a combined nested RT-qPCR technique (pan-rhabdo RT-nqPCR) dedicated to the broad detection of animal rhabdoviruses by targeting a conserved region of the polymerase (L) gene. This technique was then applied to analyze a large collection of bat samples from Africa and Europe, two different geographical regions of particular interest for rhabdoviruses, in a context of viral emergence and zoonotic transmission. For example, of the 22 different lyssaviruses discovered to date in the world, 6 have been identified in Europe and 6 others in Africa (https://ictv.global/taxonomy), whose transmission to humans or other mammals has been demonstrated [46,47]. In addition, other bat-related rhabdoviruses have also recently been discovered in these geographical regions, such as two new members of *Lendatevirus* genus with Vaprio virus in *Pipistrellus pipistrellus* bat species in Italy [34], or Kanyawara virus in pteropodid bats in Uganda [33].

Based on this screening, we were able to identify and characterize a new species among the genus *Vesiculovirus*, recognized by ICTV as *Vesiculovirus mediterranean*, which circulates in three bat species originating from the Mediterranean region. These results enhance our understanding of bat rhabdoviruses and provide a valuable tool for screening rhabdoviruses in bats and other animal reservoirs.

## 2. Materials and methods

### 2.1 Ethics approval statement

Bat samples analyzed in this study were part of biocollections previously constituted from Africa (Algeria, Central African Republic, Egypt, Guinea, Côte d'Ivoire and Morocco) and Europe (Czechia, France and Spain) in the framework of other studies. Protocols for handling and sampling individuals were covered by the specific authorities of the related countries. Sample collections from Guinea and Côte d'Ivoire were approved by the Health Ministry (2013/PFHG/05/GUI) and the Société de Développement des Forêts – SODEFOR (N°00991–16), respectively. Both were also approved by Comité Cuvier - Animal Ethics Committee of the French National Museum of Natural History (MNHN) (N°68–0009). No specific authorization was required in Central African Republic, given the unprotected status of bats in this country. Bat samples from Algeria, Egypt and Morocco were authorized by the respective Forest Ministries. Sampling of bats in Czechia were approved by the Agency for Nature Conservation and Landscape Protection of Czechia and complied with Czech Law No. 114/1992 on Nature and Landscape Protection. The collection of bat samples in Spain was authorized by the Spanish Regional Committee for Scientific Capture from Balearic Islands and Catalonia, and by the Ethical Committee of the University of Barcelona. Finally, samples were collected in France as part of the mission of rabies surveillance carried out by the National Reference Center for Rabies at Institut Pasteur, Paris.

Experimentation performed on mice was approved (agreement A75-1525, 30-07-2002) by the Ministry of National Education, Superior Education and Research under the approval number APAFIS#l5773-2018062910157376 v5, and by the Ethics Committee on Animal Experimentation (CETEA) of Institut Pasteur, under the approval numbers #15772 and #160111.

### 2.2 Samples collection

A large retrospective collection of bat samples was used for this study. This sample collection included a total of 1962 samples encompassing 423 oral swabs (21.6%), 816 blood pellet samples (41.6%) and 723 brain biopsies (36.8%), collected in 9 different countries in Europe (Czechia, France and Spain) and Africa (Algeria, Central African Republic, Egypt, Guinea, Côte d'Ivoire and Morocco) from 2007 to 2019 (Tables 1 and S1 and Fig 1). Most of the time, only one sample per bat was

**Table 1. Distribution of the bat samples tested in this study according to the country of origin, the sample type (oral swab, blood pellet and brain) and the rhabdovirus detection results.**

| Region | Country | All sample | | | Oral swab sample | | | Blood sample | | | Brain sample | |
|---|---|---|---|---|---|---|---|---|---|---|---|---|
| | | No. (%) | Pos no. (%) | Species No. | No. (%) | Pos no. (%) | Species No. | No. (%) | Pos no. (%) | Species No. | No. (%) | Species No. |
| Europe | Czechia | 234 (12) | 0 | 5 | 0 | 0 | 0 | 0 | 0 | 0 | 234 (32.3) | 5 |
| Europe | France | 71 (3.6) | 0 | 9 | 0 | 0 | 0 | 0 | 0 | 0 | 71 (9.8) | 9 |
| Europe | Spain | 373 (19) | 6 (1.6) | 5 | 341 (80.6) | 6 (1.8) | 5 | 32 (4) | 0 | 2 | 0 | 0 |
| North Africa | Algeria | 232 (11.8) | 11 (4.7) | 8 | 0 | 0 | 0 | 232 (28.4) | 11 (4.7) | 8 | 0 | 0 |
| North Africa | Egypt | 148 (7.5) | 0 | 5 | 0 | 0 | 0 | 148 (18.1) | 0 | 5 | 0 | 0 |
| North Africa | Morocco | 420 (21.4) | 6 (1.4) | 7 | 0 | 0 | 0 | 404 (49.5) | 6 (1.5) | 7 | 16 (2.2) | 3 |
| West Africa | Côte d'Ivoire | 96 (4.9) | 0 | 20 | 28 (6.6) | 0 | 13 | 0 | 0 | 0 | 68 (9.4) | 19 |
| West Africa | Guinee | 90 (4.6) | 0 | 8 | 54 (12.8) | 0 | 7 | 0 | 0 | 0 | 36 (5) | 7 |
| Central Africa | Central African Republic | 298 (15.2) | 0 | 6 | 0 | 0 | 0 | 0 | 0 | 0 | 298 (41.3) | 6 |
| **Total** | | 1962 (100) | 23 (1.2) | | 423 (100) | 6 (1.4) | | 816 (100) | 17 (2.1) | | 723 (100) | |

available for analysis, and the type of sample varied according to where and when it was collected. These bat samples belonged to 50 species of 24 genera across 8 different families, including the frugivorous bat family Pteropodidae, and the insectivorous bat families Emballonuridae, Hipposideridae, Miniopteridae, Molossidae, Rhinolophidae, Rhinopomatidae and Vespertilionidae (Tables 2 and S1). All samples from outside France were shipped to Institut Pasteur, Paris, in dry ice, in RNAlater (ThermoFisher) and/or in TRIzol (ThermoFisher). Upon reception, all samples were stored in -80°C.

After collection in the field, the quality of the samples was maintained as well as possible until their transfer to the laboratories of the countries concerned, in particular using a preservation medium (RNAlater or TRIzol) for the saliva swabs, and refrigerated coolers for the blood samples. Depending on the geographical area, the brain was obtained directly in the laboratory after transfer of the refrigerated cadaver and dissection, or in the field after dissection and use of a preservation medium (RNAlater).

The other field or laboratory samples, as well as the virus isolates used in this study for the validation method step, were obtained from the biocollection archives housed in the National Reference Center for Rabies and in the National Reference Center for Arboviruses (former), both at Institut Pasteur, Paris, France.

## 2.3 RNA extraction and cDNA synthesis

RNA extraction from oral swab samples stored in RNAlater was performed with High Pure Viral RNA Kit (Roche), according to the manufacturer's instructions, using 200 µL for each sample [28]. Brain and blood pellet samples were extracted with TRIzol using Direct-zol RNA MiniPrep (Zymo Research) according to the manufacturer's instructions. Purified RNA was stored at -80°C before use. The synthesis of cDNA was conducted with Superscript III reverse transcriptase kit (Invitrogen), following the manufacturer's specification. For each sample, a total of 8 µL RNA was used [28,48].

Negative controls were included in each extraction series, and positive and negative controls were included during the cDNA step. For most of the samples, estimation of the extracted nucleic acids was carried out by spectrophotometric assay (NanoDrop 2000, Thermo Scientific).

## 2.4 Primer design

Primer design for pan-rhabdovirus screening was performed on a set of 103 polymerase sequences from animal rhabdoviruses belonging to 14 genera within the family *Rhabdoviridae,* and available in GenBank (S2 Table and S1 Fig).

Sequence selection criteria were based on animal host (insect, fish, reptile, bird and mammal), geographical localization (all continents except Antarctica) and phylogenetic divergence/evolution (excluding rhabdovirus isolates with extremely large evolutionary distances). Sequences from this dataset, which encompassed virus isolates from 1955 to 2017, were downloaded in November 2019 and aligned using ClustalW 2.0 [49]. The most conserved regions were selected for primer design, based on the consensus sequence.

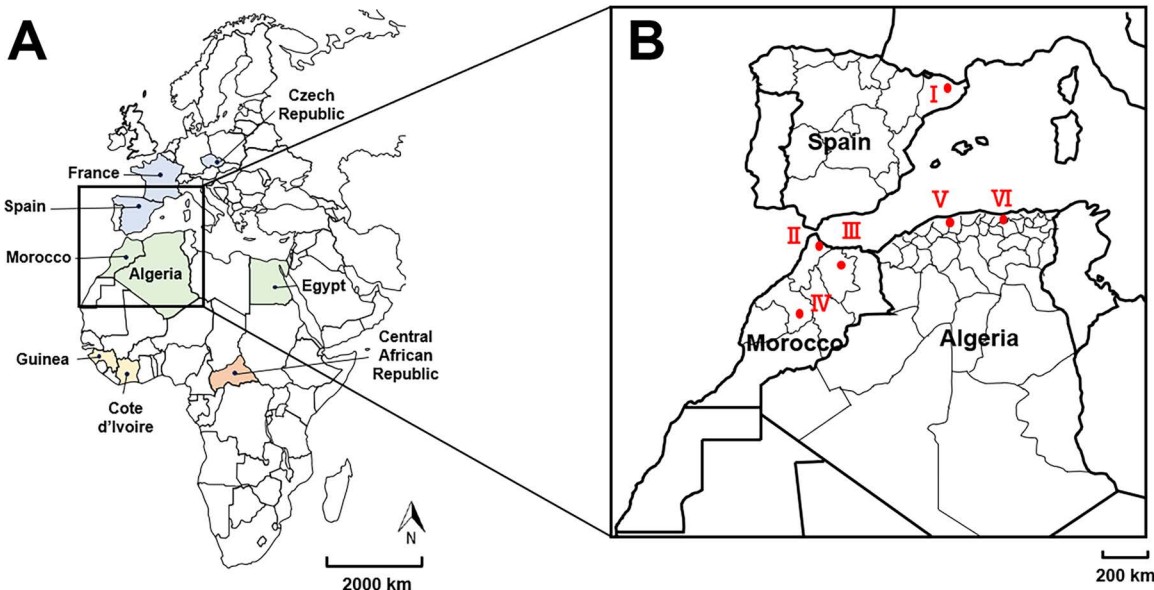

**Fig 1. Geographical distribution of retrospective bat samples analyzed in this study. A.** Geographical distribution at the country level, with samples collected from Europe (in blue) with Czechia, France and Spain; from North Africa (in green) with Algeria, Egypt and Morocco; from West Africa (in yellow) with Côte d'Ivoire and Guinea; and from central Africa (in orange) with Central African Republic. The distance scale is shown, as well as the North direction. **B.** Geographical distribution of the bat-related vesiculovirus MBV (Mediterranean bat virus) described in this study at the cave level, indicated with red dots and labelled by Roman numerals: cave I: A. Daví; cave II: Ghar-Knadel; cave III: Kef el Ghar; cave IV: Ifri N'Caid; cave V: Chrea; cave VI: Aokas. The distance scale is shown. The base layer of the world map used in Fig 1A is freely accessible at https://free-editable-world-map-for-powerpoint.en.softonic.com/download, and was manually adapted for Fig 1B.

**Table 2. Distribution of the bat samples tested in this study according to the bat families and the rhabdovirus detection results.**

| Bat family | All sample | | | Oral swab sample | | | Blood sample | | | Brain sample | |
|---|---|---|---|---|---|---|---|---|---|---|---|
| | No. (%) | Pos no. (%) | Species No. | No. (%) | Pos no. (%) | Species No. | No. (%) | Pos no. (%) | Species No. | No. (%) | Species No. |
| Pteropodidae | 514 (26.2) | 0 | 15 | 61 (14.4) | 0 | 7 | 100 (12.3) | 0 | 1 | 353 (48.8) | 14 |
| Emballonuridae | 18 (1) | 0 | 1 | 0 | 0 | 0 | 18 (2.2) | 0 | 1 | 0 | 0 |
| Hipposideridae | 40 (2) | 0 | 3 | 8 (1.9) | 0 | 3 | 0 | 0 | 0 | 32 (4.4) | 3 |
| Miniopteridae | 354 (18) | 6 (1.7) | 1 | 193 (45.6) | 6 (3.1) | 1 | 157 (19.2) | 0 | 1 | 4 (0.6) | 1 |
| Molossidae | 2 (0.1) | 0 | 1 | 0 | 0 | 0 | 0 | 0 | 0 | 2 (0.3) | 1 |
| Rhinolophidae | 183 (9.3) | 17 (9.3) | 6 | 3 (0.7) | 0 | 2 | 164 (20.1) | 17 (10.4) | 5 | 16 (2.2) | 3 |
| Rhinopomatidae | 28 (1.4) | 0 | 2 | 0 | 0 | 0 | 28 (3.4) | 0 | 2 | 0 | 0 |
| Vespertilionidae | 823 (42) | 0 | 22 | 158 (37.4) | 0 | 9 | 349 (42.8) | 0 | 7 | 316 (43.7) | 18 |
| Total | 1962 (100) | 23 (1.2) | 51 | 423 (100) | 6 (1.4) | 22 | 816 (100) | 17 (2.1) | 17 | 723 (100) | 40 |

## 2.5 Pan-rhabdovirus PCR method validation

Initial determination of the optimal concentration of these primers was performed with serial primer dilutions (1:1, 1:10 and 1:100) using the reference strain Piry virus (PIRYV, BeAn 24232, 0413BRE) (genus *Vesiculovirus*) [41,50], and led to the selection of the 200 µM concentration. The technical details concerning this validation stage can be accessed in S1 Text.

A first validation step of the different pan-rhabdovirus screening assays was conducted by comparing two previously published pan-rhabdovirus nested RT-PCR: rhabdo-screening nest conventional PCR_1 [43] and rhabdo-screening nest conventional PCR_2 [51], with five newly designed assays, including rhabdo-screening qPCR_1 to rhabdo-screening qPCR_4 and pan-rhabdo RT-nqPCR assays (S3 Table). This comparison was carried out on a limited panel (blood, brain and/or oral swab) of negative ($n = 10$) and positive ($n = 10$) samples, including representative members of the genera *Ephemerovirus*, *Ledantevirus*, *Sunrhavirus* and *Vesiculovirus* (S4 Table).

The sensitivity and specificity parameters of the selected pan-rhabdo molecular screening method (pan-rhabdo RT-nqPCR) was further validated using a large panel of positive and negative samples. Positive samples included representative virus members ($n = 60$) of different genera ($n = 8$) of the family *Rhabdoviridae*, and corresponded to laboratory or field samples (brain, blood or cell culture) from various animal species (*e.g.,* bat, bovine, dog, fox, mouse) (S5 Table). Negative samples encompassed viruses ($n = 18$) from other viral families ($n = 9$), as well as uninfected biological samples ($n = 7$) from different animal species (bat, dog, fox, monkey) and tissue types (brain, cell line) (S6 Table). Based on this validation step, an analytic workflow was designed and applied for bat sample screening and result interpretation.

## 2.6 Validation of bat positive samples obtained with the pan-rhabdo RT-nqPCR assay

Bat samples were considered positive for rhabdovirus detection when the characteristics of the dissociation curve (shape and melting temperature) observed with the amplicons were similar to those obtained with positive controls. Confirmation was done after Sanger sequencing of the amplicons and analysis of the sequences using Sequencher 5.2.4 (Gene Codes Corporation). In case of ambiguous results, amplicons were cloned using TOPO TA Cloning Kit (Invitrogen) according to the manufacturer's instructions. In this case, a total of 5–10 clones for each sample were selected for Sanger sequencing, and sequence contigs were analyzed by BLASTn and BLASTx using the NCBI database.

## 2.7 Virus genome sequencing

Nearly complete genome sequences of bat-related rhabdoviruses were obtained using Illumina next generation sequencing (NGS) technology, as previously described [28,48]. More technical details concerning can be accessed in S1 Text.

Contigs were obtained by combining *de novo* assembly and mapping (both with CLC Assembly Cell, Qiagen) using a dedicated workflow implemented on Galaxy@Pasteur (a Galaxy platform hosted by Institut Pasteur) [19,28,48,52]. The contigs were then assembled and manually edited to produce the final consensus genome sequences using Sequencher. The quality and accuracy of the final genome sequences were validated after a final mapping step of the original cleaned reads and visualized using Tablet [53].

Gaps or ambiguous nucleotide positions were corrected by (nested) PCR using the TaKaRa EX Taq (TaKaRa) according to the manufacturer's instructions, using specific primers designed from the genome sequences obtained by NGS (S7 Table). Amplicons were submitted to Sanger sequencing and sequence correction was done using Sequencher.

Genome sequences were deposited in NCBI GenBank under the accession numbers MW557328-MW557343.

## 2.8 Genome analysis

Putative coding regions of the genome sequences were determined using Sequencher, and putative accessory genes were evaluated after comparison with similar rhabdovirus genome sequences available in GenBank.

Phylogenetic analysis of the bat vesiculoviruses and other members of the genus *Vesiculovirus* was conducted using a nucleotide sequence dataset downloaded from NCBI GenBank (S8 Table). For each virus, complete genome nucleotide sequences were aligned using ClustalW (version 2.0) [49] and manually checked for accuracy. Phylogenetic trees were constructed with MEGA (version 7.0) [54] and PhyML (version 3.0) [55] using the Maximum Likelihood method, with GTR+G+I model and with 1000 bootstrap replicates. Pairwise sequence identities were performed using MEGA.

### 2.9 Virus isolation

Virus isolation was attempted in newborn suckling BALB/c mice (3 days old) (Charles River laboratories). For each positive sample, 20–50 µL of lysed whole blood were diluted into 100 µL sterile PBS, gently mixed and centrifuged at 5000 g for 10 min at 4°C. The supernatant (5–10 µL) was inoculated intracerebrally into 4–5 newborn mice per sample. Inoculated animals were monitored daily and euthanized on day 5 or 7 after inoculation. The presence of the virus was tested on the brain after extraction of total RNA and detection by pan-rhabdo RT-nqPCR, as previously described.

## 3. Results

### 3.1 Design and selection of the primers for the broad-spectrum detection of animal rhabdoviruses

A dataset of 103 polymerase nucleotide sequences from rhabdovirus species belonging to 14 different genera was selected for multiple sequence alignment (S2 Table and S1 Fig). After analysis, a highly conserved region of the polymerase gene (positions 1600 and 2207 according to the sequence reference Drosophila obscura Sigmavirus, NCBI Accession NC_022580) was identified and selected for primer design. Of the 7 different primers designed and associated with 5 PCR systems (S3 Table), an initial sensitivity and specificity validation stage resulted in the selection of the SYBR Green-based nested qPCR system (pan-rhabdo RT-nqPCR) (S4 Table). Indeed, this technique provided 100% of specificity ($n = 10$) and sensitivity ($n = 10$), compared the two previously published methods, with rhabdo-screening nest conventional PCR_1 (0% sensitivity, $n = 4$) [43] and rhabdo-screening nest conventional PCR_2 (75% sensitivity, $n = 4$) [51].

### 3.2 Validation of the pan-rhabdo RT-nqPCR assay

Based on the preliminary validation step, a standardized process of analysis was established, using two main parameters: Cp value and dissociation curve (Tm value and shape) (S2 Fig). The sensitivity of this assay was evaluated using this process and a panel of 60 positive samples representative of 38 different rhabdovirus species from 8 different genera (S5 Table). All these samples were detected using the pan-rhabdo RT-nqPCR assay, providing 100% sensitivity. The specificity of this assay was also evaluated on a panel of 18 viruses belonging to 9 different families other than *Rhabdoviridae*, and on 7 non-infected samples from 5 different animal species (S6 Table). All these samples were not detected by the pan-rhabdo RT-nqPCR assay, exhibiting 100% specificity.

### 3.3 Bat sample screening with the pan-rhabdo RT-nqPCR assay

Among the 1962 bat sample tested, 23 (1.2%) tested positive for rhabdovirus. All came from 3 different bat species, including 6 samples from *Miniopterus schreibersii*, 2 from *Rhinolophus euryale* and 15 from *Rhinolophus ferrumequinum*, leading to a prevalence of 1.7% ($n = 354$), 4.9% ($n = 41$) and 11.9% ($n = 126$), respectively, considering the total number of samples tested for each of these species (Table 3). All positive bats originated from the Mediterranean region, with 6 bats from Spain, 11 from Algeria and 6 from Morocco, resulting in a prevalence per country of 1.6% ($n = 373$), 4.7% ($n = 232$) and 1.4% ($n = 420$), respectively. None of the brain samples tested positive, while rhabdovirus detection by the pan-rhabdo RT-nqPCR assay was obtained for 6 oral swab samples (1.4%, $n = 423$) and 17 blood samples (2%, $n = 816$), collected in 2008, 2009 and 2012 (Table 3). Sanger sequencing of the amplicons confirmed the detection of rhabdoviruses, probably related to the genus *Vesiculovirus*.

**Table 3. Results of the rhabdovirus detection with the pan-rhabdo RT-nqPCR for a collection of retrospective bat samples (blood, brain, oral swab) according to bat species and geographical location.**

| Bat | | No. positive/ No. tested (%) | | | | | | | | | |
|---|---|---|---|---|---|---|---|---|---|---|---|
| Family | Species | Europe | | | North Africa | | | West Africa | | Central Africa | Total (%) |
| | | Czechia | France | Spain | Algeria | Morocco | Egypt | Guinea | Côte d'Ivoire | Central Africa Republic | |
| Pteropodidae | *Eidolon helvum* | | | | | | | | | 0/222 (0) | 0/222 (0) |
| | *Epomophorus gambianus* | | | | | | | 0/8 (0) | 0/2 (0) | 0/2 (0) | 0/12 (0) |
| | *Epomops buettikoferi* | | | | | | | | 0/5 (0) | | 0/5 (0) |
| | *Epomops franqueti* | | | | | | | | 0/1 (0) | 0/2 (0) | 0/3 (0) |
| | *Epomops sp.* | | | | | | | 0/4 (0) | 0/1 (0) | | 0/5 (0) |
| | *Hypsignathus gambianus* | | | | | | | | | 0/1 (0) | 0/1 (0) |
| | *Hypsignathus monstrosus* | | | | | | | | 0/2 (0) | 0/3 (0) | 0/5 (0) |
| | *Megaloglossus azagnyi* | | | | | | | | 0/1 (0) | | 0/1 (0) |
| | *Micropteropus pusillus* | | | | | | | 0/65 (0) | 0/7 (0) | 0/68 (0) | 0/140 |
| | *Myonycteris leptodon* | | | | | | | | 0/7 (0) | | 0/7 (0) |
| | *Myonycteris torquata* | | | | | | | 0/4 (0) | | | 0/4 (0) |
| | *Nanonycteris veldkampii* | | | | | | | 0/1 (0) | 0/4 (0) | | 0/5 (0) |
| | *Rousettus aegyptiacus* | | | | | | 0/100 (0) | | | | 0/100 (0) |
| | *Scotonycteris zenkeri* | | | | | | | | 0/4 (0) | | 0/4 (0) |
| Emballonuridae | *Taphozous nudiventris* | | | | | | 0/18 (0) | | | | 0/18 (0) |
| Hipposideridae | *Hipposideros caffer* | | | | | | | | 0/33 (0) | | 0/33 (0) |
| | *Hipposideros ruber* | | | | | | | 0/1 (0) | 0/3 (0) | | 0/4 (0) |
| | *Hipposideros sp.* | | | | | | | | 0/3 (0) | | 0/3 (0) |
| Miniopteridae | *Miniopterus schreibersii* | | | 6/213 (2.8) | 0/56 (0) | 0/85 (0) | | | | | 6/354 (1.7) |
| Molossidae | *Molossus molossus* | | 0/2 (0) | | | | | | | | 0/2 (0) |
| Rhinolophidae | *Rhinolophus alcyone* | | | | | | | 0/5 (0) | | | 0/5 (0) |
| | *Rhinolophus blasii* | | | | 0/1 (0) | | | | | | 0/1 (0) |
| | *Rhinolophus euryale* | | | | 1/31 (3.2) | 1/10 (10) | | | | | 2/41 (4.9) |
| | *Rhinolophus ferrumequinum* | | | | 10/46 (21.7) | 5/80 (6.2) | | | | | 15/126 (11.9) |
| | *Rhinolophus hipposideros* | | | | 0/1 (0) | | | | | | 0/1 (0) |
| | *Rhinolophus sp.* | | | | | 0/6 (0) | | | 0/3 (0) | | 0/9 (0) |
| Rhinopomatidae | *Rhinopoma hardwickii* | | | | | | 0/7 (0) | | | | 0/7 (0) |
| | *Rhinopoma microphyllum* | | | | | | 0/21 (0) | | | | 0/21 (0) |

*(Continued)*

**Table 3.** (Continued)

| Bat | | No. positive/ No. tested (%) | | | | | | | | | |
|---|---|---|---|---|---|---|---|---|---|---|---|
| Family | Species | Europe | | | North Africa | | | West Africa | | Central Africa | Total (%) |
| | | Czechia | France | Spain | Algeria | Morocco | Egypt | Guinea | Côte d'Ivoire | Central Africa Republic | |
| Vespertilionidae | *Eptesicus isabellinus* | | | | | 0/3 (0) | | | | | 0/3 (0) |
| | *Eptesicus serotinus* | | 0/32 (0) | | | | | | | | 0/32 (0) |
| | *Myotis blythii* | | | 0/13 (0) | | | | | | | 0/13 (0) |
| | *Myotis capaccinii* | | | 0/7 (0) | 0/7 (0) | | | | | | 0/14 (0) |
| | *Myotis dasycneme* | 0/1 (0) | | | | | | | | | 0/1 (0) |
| | *Myotis emarginatus* | | 0/3 (0) | | 0/27 (0) | | | | | | 0/30 (0) |
| | *Myotis escalerai* | | | 0/9 (0) | | 0/1 (0) | | | | | 0/10 (0) |
| | *Myotis myotis* | 0/188 | | 0/131 (0) | | | | | | | 0/319 (0) |
| | *Myotis mystacinus* | | 0/12 (0) | | | | | | | | 0/12 (0) |
| | *Myotis nattereri* | | 0/1 (0) | | | | | | | | 0/1 (0) |
| | *Myotis punicus* | | | | 0/63 (0) | 0/235 (0) | | | | | 0/298 (0) |
| | *Afronycteris nana* | | | | | | | | 0/4 (0) | | 0/4 (0) |
| | *Nyctalus noctula* | 0/39 (0) | 0/9 (0) | | | | | | | | 0/48 (0) |
| | *Nyctalus leisleri* | | 0/1 (0) | | | | | | | | 0/1 (0) |
| | *Nycteris grandis* | | | | | | | | 0/2 (0) | | 0/2 (0) |
| | *Nycteris hispida* | | | | | | | 0/2 (0) | | | 0/2 (0) |
| | *Nycteris thebaica* | | | | | | | | 0/4 (0) | | 0/4 (0) |
| | *Nycteris sp.* | | | | | | | | 0/2 (0) | | 0/2 (0) |
| | *Pipistrellus kuhlii* | | 0/5 (0) | | | | 0/2 (0) | | | | 0/7 (0) |
| | *Pipistrellus pipistrellus* | 0/1 (0) | 0/6 (0) | | | | | | | | 0/7 (0) |
| | *Scotophilus leucogaster* | | | | | | | | 0/8 (0) | | 0/8 (0) |
| | *Vespertilio murinus* | 0/5 (0) | | | | | | | | | 0/5 (0) |
| Total (%) | | 0/234 (0) | 0/71 (0) | 6/373 (1.6) | 11/232 (4.7) | 6/420 (1.4) | 0/148 (0) | 0/90 (0) | 0/96 (0) | 0/298 (0) | 23/1962 (1.2) |

All *Miniopterus schreibersii* positive samples were found in oral swabs in Spain in the Catalonian cave I (A. Daví) in 2012, corresponding to a prevalence of 33.3% of the total oral swab number tested in this cave ($n = 18$) (Table 4 and Fig 1). The two positive *Rhinolophus euryale* samples were detected in blood pellet from cave III and cave VI in Morocco and Algeria in 2008, respectively. The positive rate in blood for this bat species was 25% ($n = 4$) and 3.6 ($n = 28$) for caves III and VI, respectively. Finally, the highest number of positive samples ($n = 15$) was found in *Rhinolophus ferrumequinum* bat blood pellets collected in two caves (caves II and IV) in Morocco and in two caves (caves V and VI) in Algeria during the period 2008–2009 (Table 4 and Fig 1). The overall prevalence in each of these caves for this bat species was 7.5% ($n = 40$), 14.3% ($n = 14$), 22.2% ($n = 36$) and 40% ($n = 5$) for caves II, IV, V and VI, respectively. Interestingly, samples from cave V in Algeria were collected in 2008 and 2009, with a prevalence of 14.3% ($n = 7$) and 24.1% ($n = 29$), respectively, highlighting the persistence of the infection over the two-year period.

**Table 4. Results of the rate of rhabdovirus detection with the pan-rhabdo RT-nqPCR in oral swab and brain samples collected from different bat species and in selected caves from Algeria, Morocco and Spain.**

| Collection year | No. positive/ No. tested (%) | | | | | | | Total (%) |
|---|---|---|---|---|---|---|---|---|
| | *Miniopterus schreibersii* | *Rhinolophus euryale* | | *Rhinolophus ferrumequinum* | | | | |
| | Cave[a] I (Spain) | Cave III (Morocco) | Cave VI (Algeria) | Cave II (Morocco) | Cave IV (Morocco) | Cave V (Algeria) | Cave VI (Algeria) | |
| 2008 | | 1/4 (25) | 1/28 (3.6) | 3/40 (7.5) | | 1/7 (14.3) | | 6/79 (7.6) |
| 2009 | | | | | 2/14 (14.3) | 7/29 (24.1) | 2/5 (40) | 11/48 (22.9) |
| 2012 | 6/18 (33.3) | | | | | | | 6/18 (33.3) |
| Total (%) | 6/18 (33.3) | 1/4 (25) | 1/28 (3.6) | 3/40 (7.5) | 2/14 (14.3) | 8/36 (22.2) | 2/5 (40) | 23/145 (15.9) |

[a]The identification of the caves is as follows: cave I: A. Daví, cave II: Ghar-Knadel, cave III: Kef el Ghar, cave IV: Ifri N'Caid, cave V: Chrea, cave VI: Aokas.

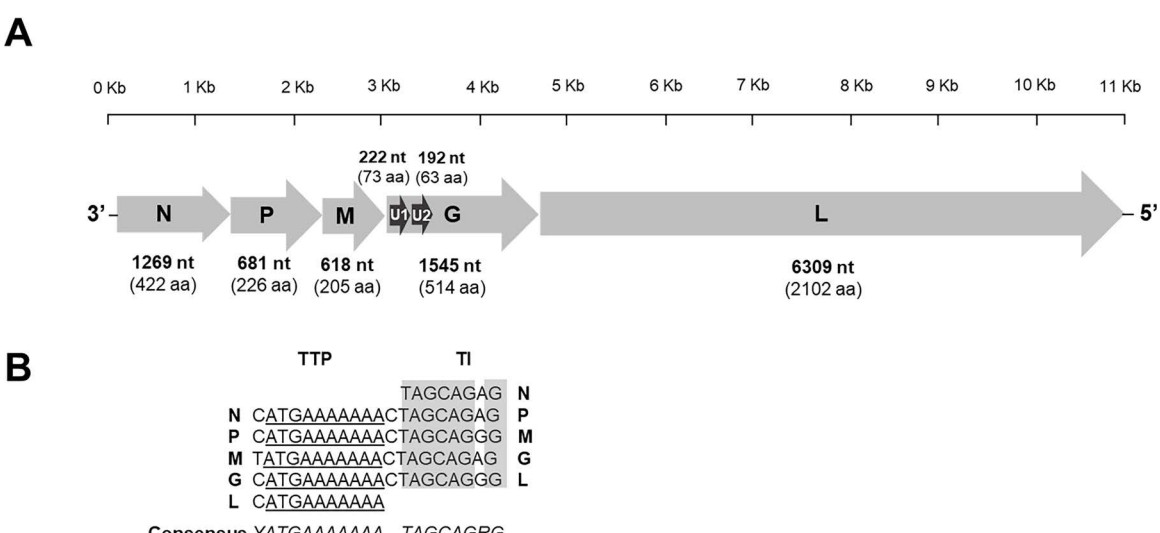

**Fig 2. Analysis of the genome of the bat rhabdovirus detected by the pan-rhabdo RT-nqPCR. A.** Schematic organization of the genome. Grey arrows represent the five canonical open reading frames (ORFs) with the nucleoprotein (N), the phosphoprotein (P), the matrix protein (M), the glycoprotein (G) and the RNA polymerase RNA dependant (L). Black arrows indicate the position of putative additional ORFs. The nucleotide and amino acid lengths of each ORF are indicated. **B.** Description of the transcription initiation (TI) and the transcription termination/polyadenylation (TTP) signal sequences of the five canonical ORFs (N, P, M, G and L) indicated in left or in right, respectively. Conserved motifs are indicated in grey for TI or underlined for TTP. The consensus sequences are indicated in italic, with Y = C/T and R = A/G.

### 3.4 Complete genome analysis of bat rhabdoviruses

Between 3–14 million raw reads were obtained per sample (around 5 million reads on average) for the 23 positive samples. Rhabdovirus-specific reads or contigs were obtained for all of them, but only 16 samples exhibited a large genome contigs after analysis (S9 Table). The average coverage for each of them ranged from 1.58x to 2472x. Remaining gaps and low-coverage regions were resolved by specific PCR or nested-PCR, following by Sanger sequencing. After a final verification step carried out by mapping using the completed sequence consensus as a reference, 16 nearly complete genome sequences were obtained, ranging in length from 10,914–11,093 nt (excluding the 3′ Leader and 5′ Trailer sequences) (Table 5).

**Table 5. Read coverage of the genome sequences of the 16 Mediterranean bat virus (MBV) isolates, obtained after a last mapping step using NGS data.**

| Isolate | Country | Sample type | Raw reads (No. x10⁶) | Cleaned reads (No. x10⁶) | Mapped reads (No.) | Sequence size (nt) | Average coverage (X) | Length sequence coverage (%) |
|---------|---------|-------------|----------------------|--------------------------|--------------------|--------------------|----------------------|------------------------------|
| A08011 | Algeria | Blood | 3.798 | 2.662 | 1877 | 10,914 | 25.35 | 100 |
| A08065 | Algeria | Blood | 3.842 | 2.735 | 1156 | 10,938 | 15.71 | 100 |
| A09061 | Algeria | Blood | 5.152 | 4.057 | 189,052 | 11,092 | 2472.19 | 100 |
| A09097 | Algeria | Blood | 6.270 | 4.734 | 11,433 | 11,092 | 151.80 | 100 |
| A09145 | Algeria | Blood | 3.580 | 2.596 | 57,269 | 11,093 | 758.85 | 100 |
| A09151 | Algeria | Blood | 5.555 | 4.141 | 15,805 | 11,093 | 210.67 | 100 |
| A09153 | Algeria | Blood | 3.415 | 2.447 | 19,037 | 11,065 | 251.75 | 100 |
| A09181 | Algeria | Blood | 4.662 | 3.452 | 128,719 | 11,093 | 1686.87 | 100 |
| A09193 | Algeria | Blood | 3.398 | 2.543 | 4658 | 11,093 | 61.87 | 100 |
| A09197 | Algeria | Blood | 6.372 | 4.938 | 2464 | 11,011 | 33.09 | 100 |
| M08013 | Morocco | Blood | 5.570 | 3.889 | 3233 | 10,980 | 43.46 | 100 |
| M08017 | Morocco | Blood | 3.854 | 2.743 | 3740 | 10,932 | 50.79 | 100 |
| M08051 | Morocco | Blood | 5.391 | 3.860 | 1085 | 10,975 | 14.65 | 100 |
| M09005 | Morocco | Blood | 4.787 | 3.793 | 4278 | 10,920 | 58.21 | 100 |
| M09009 | Morocco | Blood | 5.006 | 3.893 | 38,266 | 11,039 | 513.67 | 100 |
| 2012096 | Spain | Oral swab | 14.188 | 9.189 | 122 | 11,068 | 1.58 | 52.2 |

The 16 genome sequences exhibited a common and typical rhabdovirus organization, consisting of five canonical genes encoding, in the following order, the nucleoprotein (N) (422 aa, 1269 nt), the phosphoprotein (P) (226 aa, 681 nt), the matrix protein (M) (205 aa, 618 nt), the glycoprotein (G) (514 aa, 1545 nt), and the RNA polymerase (L) (2102 aa, 6309 nt) (Fig 2A). For 4 genome sequences (A08011, A08065, M08017 and M09005), the N coding sequence remained incomplete after sequencing, with 40–60 missing nucleotides after the start codon. The transcription initiation (TI) signal was highly conserved, with the TAGCAGRG sequence (R=A/G), while the transcription termination/polyadenylation signal (TTP) was the consensus sequence YATG(A)$_7$ (Y=C/T) (Fig 2B). In addition, two potential coding sequences representing the putative accessory genes U1 (73 aa, 222 nt) and U2 (63 aa, 192 nt) were identified in the G gene (Fig 2A).

After concatenation of the coding sequences for each genome and pairwise comparison, the sequences showed high nucleotide identities, ranging from 97.3% to 99.9% (S10 Table). Amino acid identity deduced for each of the five canonical protein was also high between the 16 genome sequences of MBV isolates, with identities of 100% for N and M proteins, between 98.6% and 100% for the P and G proteins, and ≥ 99% for the L proteins, suggesting that they were all classified in a species provisionally named Mediterranean bat virus (MBV) (S11 Table).

Finally, BLASTn and BLASTx analyses performed out on the 16 genome sequences demonstrated that all these viruses were closely related to bat rhabdoviruses belonging to the genus *Vesiculovirus*, and more specifically to the recently described Chinese bat-related rhabdoviruses, such as Yinshui bat virus (GenBank number MN607597) or Qiong-zhong bat virus (GenBank number MN607593), with approximately 60% nucleotide identity.

Similar sequence comparisons and BLAST analyses conducted on the short contigs (< 300 nt) obtained after NGS with the 7 other positive bat samples demonstrated that each was closely related to MBV.

### 3.5 Phylogenetic analysis of the Mediterranean bat virus

Maximum-likelihood phylogenetic analysis was performed on the 16 nucleotide sequences of the MBV genome and representative members of the genus *Vesiculovirus*. Phylogeny confirmed that MBV belonged to this genus *Vesiculovirus*.

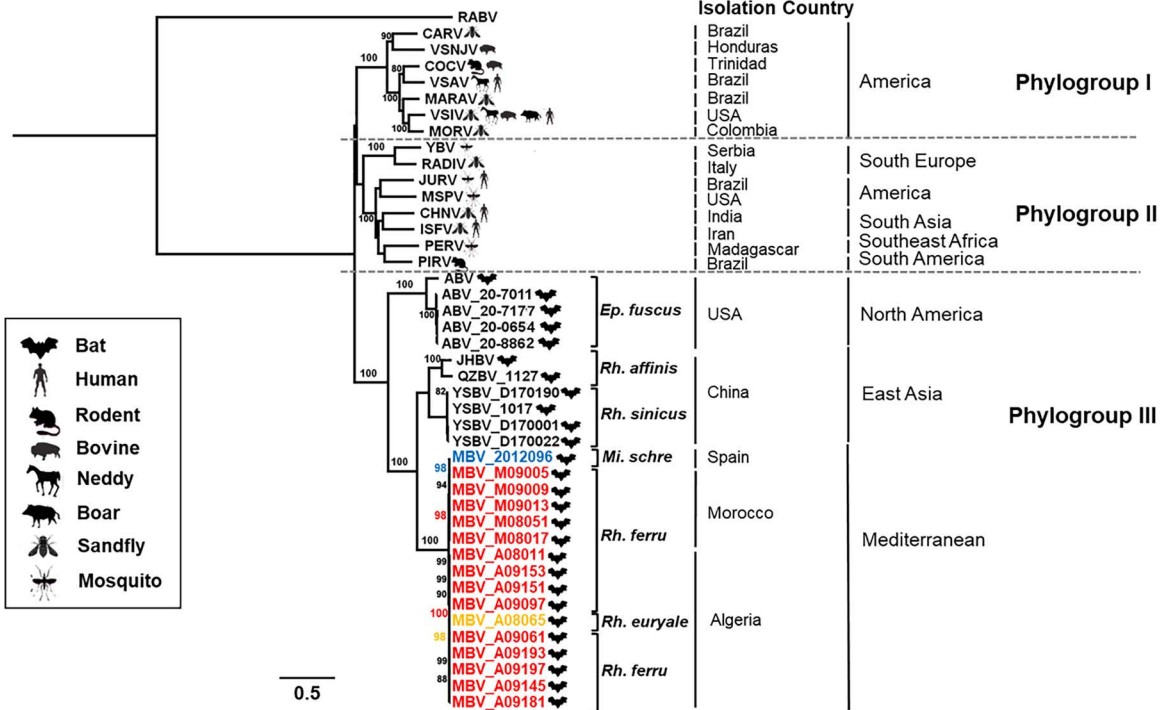

**Fig 3. Phylogenetic classification of all members of the genus *Vesiculovirus,* including the Mediterranean bat virus (MBV) detected in this study.** A maximum likelihood phylogenetic tree was done with PhyML3.0 on the nucleotide of complete genome, using the GTR+G+I model and 1000 bootstrap replicates. The main animal reservoirs for each virus are indicated by specific cartoons and the 16 bat rhabdovirus isolates of MBV described in this study are indicated in red when obtained from *Rhinolophus ferrumequinum*, in orange when obtained from *Rhinolophus euryale*, and in blue for the isolate collected from *Miniopterus schreibersii* in Spanish (the only one detected in oral swab compared to others detected in blood). The bat species information was represented by abbreviation on the figure (*Ep.fuscus: Eptesicus fuscus, Rh.affinis: Rhinolophus affinis, Rh.sinicus: Rhinolophus sinicus, Mi.schre: Miniopterus schreibersii, Rh.ferru: Rhinolophus ferrumequinum, Rh.euryale: Rhinolophus euryale*), and the isolation country for each virus was presented in the right of the illustration. All bootstrap proportion values (BSP) > 80% are specified. Scale bar indicates nucleotide substitutions per site. The classical rabies virus (RABV) was included for out clade in the phylogenetic analysis.

Within this genus, three phylogroups were identified, based on the genetic clustering, geographical and/or host origins, with the phylogroup I comprising various viruses from America whereas the phylogroup II was cosmopolitan, encompassing viruses from Africa, American, Asia and Europe (Fig 3). Interestingly, all bat-related vesiculoviruses were clustered into the same phylogroup III. Within this phylogroup, a geographical clustering was observed, with three main groups: one encompassing bat vesiculoviruses obtained from *Eptesicus fuscus* in North America (with American bat vesiculovirus or ABV), the second grouping Asian (and more particularly Chinese) bat vesiculoviruses with two distinct branches according to the host bat species (*Rhinolophus affinis* or *Rhinolophus sinicus*), and the last one clustering together all the MBV isolates from the Mediterranean region, regardless of the host bat species (Fig 3).

### 3.6  Genetic diversity of the Mediterranean bat virus

A phylogenetic clustering was observed between countries of origin, and more specifically between caves. For example, Algerian MBV isolates formed a distinct and well supported group, subdivided into subclusters according to the cave origins, *i.e.* cave VI, V and VI (Fig 4). A specific phylogroup was also identified for the Moroccan strains from Cave II. Interestingly, the Moroccan samples from Cave IV clustered with the single Spanish MBV isolate, which still appears to be genetically distinct.

Within the genus *Vesiculovirus*, the highest percentage of deducted amino-acid identity for each of the 5 coding regions (N, P, M, G and L) was observed for the species *Vesiculovirus yinshui* (with isolate YSBV 1017), following with the species *Vesiculovirus jinghong* (with isolates QZBV 1127 and JHBV) (S12 Table). For the first and more closely related species YSBV, these values were 75.5%, 21.1%, 61.5%, 62.4% and 67% for the N, P, M, G and L proteins, respectively (S12 Table). These results suggest that MBV virus represents a novel vesiculovirus species.

Among the deduced P, G and L proteins, the overall number of mutations, regardless of geographical location, was 28, with 2 for P, 7 for G and 19 for L proteins. As expected, the number of amino acid mutations was lower within MBV belonging to the same geographical cluster. A total of 8 mutations were found for the 10 isolates from Algeria, including one in the G protein and 7 in the L protein, while 7 mutations were observed in the 5 viruses from Morocco, including 2 in the P protein, 2 in the G protein 3 in the L protein (S3 Fig). As expected from the phylogenetic analysis, the Spanish strain was closer to the Moroccan strains, with only 1 mutation in the G protein and 4 in the L protein using the M09009 strain from the cave IV.

### 3.7 Virus isolation

Virus isolation was performed after intracranial inoculation of newborn BALB/c mice (3 days old) with remaining blood pellets. After 7 days of observation, no clinical symptoms nor mortality were observed. Results obtained with the pan-rhabdo-RT-qPCR after total RNA extraction from the inoculated mice brains remained negative.

### 4. Discussion

To further explore the diversity and the prevalence of rhabdoviruses in bats, we have successfully developed and validated a combined nested RT-qPCR technique (pan-rhabdo RT-nqPCR) dedicated to the broad detection of animal rhabdoviruses, targeting a conserved region in the polymerase gene. Comparison of this technique with similar previously published generic methods suggested a higher sensitivity, although the sample panel used was limited.

When applied to a large collection of retrospective bat samples from different countries, the overall prevalence was low, with 23 positive samples (1.2%) in total, including 17 blood pellet (2.1%) and 6 oral swab (1.4%) samples, all originated from Algeria (4.7%), Morocco (1.4%) and Spain (1.6%). Only three of the 51 different bat species tested were found positive: *Miniopterus schreibersii* ($n=6$), *Rhinolophus euryale* ($n=2$) and *Rhinolophus ferrumequinum* ($n=15$). Interestingly, all the positive bats were collected between 2008 and 2012 in the Mediterranean basin, with Algeria ($n=11$) and Morocco ($n=6$) for the two *Rhinolophus* species, and Spain ($n=6$) for the *Miniopterus* species. Although low, these prevalence rates are higher than those found in other studies. For example, Aznar-Lopez et al obtained 0.7% (10/1488) of positivity using the nested DimLis PCR RT-PCR method in oropharyngeal samples collected from five of the 27 bat species sampled throughout Spain between 2004 and 2010 [43]. In this study, the bat species found positive for rhabdoviruses were *Eptesicus isabellinus*, *Hypsugo savii*, *Miniopterus schreibersii*, *Plecotus auritus* and *Rhinolophus ferrumequinum*. Another study carried out in the *Desmodus rotundus* bat (vampire bat) in Guatemala using a generic hemi-nested PCR based on the primers PVO3/PVO4/ PVOnstF provided a positivity rate of 0.25% (1/396) after screening different samples (blood clot, serum, fecal or oral swab and urine) [51]. The higher prevalence found in our study could be linked to greater methodological sensitivity, but the different technical approaches used between studies, as well as the differences between the different bat populations tested, mean that a robust comparison cannot be made.

Although high, we cannot completely exclude a potential negative impact of the quality of some samples on the results of this prevalence, despite the implementation of measures aimed at maximizing the conservation of samples taken in the field (quality of sampling, use of a conservation medium such as RNAlater or TRIzol, conditions of transport and refrigerated storage, etc.).

After Sanger sequencing, all the 23 positive samples exhibited the presence of the same rhabdovirus isolate, named Mediterranean bat virus (MBV), and belonging to the genus *Vesiculovirus*. The almost complete genome sequence was

retrieved for 16 samples (1 from Spain, 5 from Morocco and 10 from Algeria), whereas only few viral reads were obtained from the 7 other samples, probably due to low viral load or greater degradation. The genome sequence was characterized by a classical vesiculovirus organization with the presence of the five canonical genes (N, P, M, G, L). For each of these proteins, amino acid identities were high (over 99%), demonstrating the close genetic relationships between these isolates. Within the genus *Vesiculovirus*, the genetic comparison between the other members revealed that Mediterranean bat virus can be considered as a new species. Indeed, MBV exhibited amino acid divergences of 33%, 24.5% and 37.6% for the L, N and G proteins, respectively, with the most closely related vesiculovirus being the species *Vesiculovirus yinshui*. These genetic divergence results were in accordance with the ICTV demarcation criteria (minimum amino acid sequence divergence of 20%, 10% and 15% in L, N and G proteins, respectively) to validate the creation of a new virus species, and the *Vesiculovirus mediterranean* species was ratified in March 2022, with MBV representing the prototype virus [18].

Within the genus *Vesiculovirus*, MBV isolates clustered in a distinct group within phylogroup III, the latter also encompassing other bat-related vesiculoviruses, with one group bringing together American bat vesiculoviruses (ABV) collected from *Eptesicus fuscus* in North America [26] and the other one composed of Chinese bat-related vesiculoviruses with Yinshui bat virus (YSBV) from *Rhinolophus sinicus*, and with Qiongzhong bat virus (QZBV) or Jinghong bat virus (JHBV), both obtained from *Rhinolophus affinis* [27,28]. These results demonstrated that specific bat-related vesiculoviruses are actively circulating among various bat species worldwide.

The phylogenetic analysis of the different genome sequences of MBV isolates also revealed a geographical demarcation, with a distinct cluster of Algerian isolates, and a second cluster encompassing both the single Spanish strain and

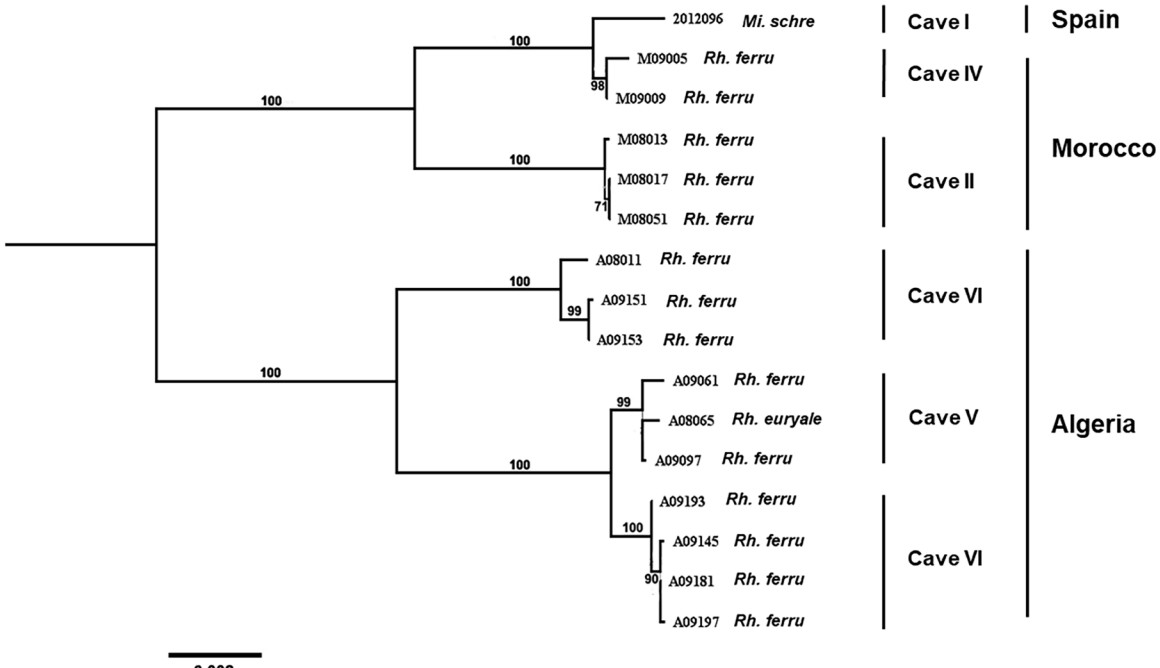

**Fig 4. Details of the phylogenetic relationship between the 16 isolates of Mediterranean bat virus (MBV).** A maximum likelihood phylogenetic tree was done with PhyML3.0 on the nucleotide of complete genome, using the GTR + G + I model and with 1000 bootstrap replicates. All bootstrap proportion values (BSP) > 70% are specified. Scale bar indicates nucleotide substitutions per site. The tree was mid-point rooted for clarity only. The identification of the bat species (*Mi.schre: Miniopterus schreibersii, Rh.ferru: Rhinolophus ferrumequinum* and *Rh.euryale: Rhinolophus euryale*), of the caves and of the country of origin are indicated.

the isolates from Morocco. In the latter cluster, the Spanish isolate was more genetically related to the Moroccan ones, suggesting that the Spanish strain may have originated from Moroccan strains. Looking more closely at the scale of the caves, this demarcation was even more striking, since all the isolates in a single cave were distinct from those in other caves, even within the same country and during similar time period (2008–2009 for the Algerian and Moroccan strains). Interestingly, a limited number of mutations was observed between the different MBV isolates, especially in the deduced P, G and L proteins, and was related to geographical cluster. So far, the impact of these mutations is unknown, but could affect cell recognition (mutation in G) or the replication complex (mutations in P and L proteins).

Unlike the other bat vesiculoviruses, MBV is not restricted to one bat species, being able to infect at least three bat species to date, especially those of the genus *Rhinolophus*. This ability to infect different species of mammals is also found in most other arthropod-borne vesiculoviruses, infecting birds and reptiles and different species of mammals, especially livestock, in endemic areas. Although populations are small in our study, there does not appear to be any viral genetic demarcation according to host bat species, at least within the genus *Rhinolophus*. Indeed, the Algerian MBV A08065 isolate and both A09061 and A09097 isolates obtained from the same cave (cave V) from *Rhinolophus ferrumequinum* and *Rhinolophus euryale*, respectively, are genetically almost identical.

These three positive bat species often share the same roost. In Morocco, caves III and IV host various mono-species bat colonies, with *Miniopterus schreibersii*, *Myotis emarginatus*, *Myotis punicus*, *Rhinolophus ferrumequinum* and *Rhinolophus euryale*. However, in these roosts, these species were often observed in close contact, which may facilitate the spread and circulation of MBV between individuals of different species. Furthermore, *Miniopterus schreibersii* is the only species capable of relatively long seasonal movements, sometimes covering distances of over 300 kilometers. It migrates between winter and summer roosts, potentially contributing to the dispersal of MBV in northern Africa, although little is known about its migration patterns in this region.

The migration of *Miniopterus schreibersii* is most documented in the Catalonia region of Spain, where cave I is located. This roost hosts a large wintering bat colony comprising 17,000 *Miniopterus schreibersii* and 150 *Rhinolophus ferrumequinum*. Towards the end of winter, *Miniopterus schreibersii* migrate to spring roosts and then to summer sites [56]. Thus, the bats in cave I disperse over a vast territory, encompassing a significant part of Catalonia and southeastern France, potentially contributing to the dispersal and maintenance of MBV circulation. In addition, studies of other rhabdoviruses have demonstrated the importance of *Miniopterus schreibersii* in maintaining lyssavirus in colonies of other bat species [57,58]. It is possible that *Miniopterus schreibersii* plays a similar role with MBV, but further research is needed to confirm this hypothesis. Seroprevalence determination is a useful parameter to determine the level of bat exposure, and to try to model the dynamics of circulation of this virus in bat populations, particularly when the positivity rate of virus detection remains low. Indeed, such approaches have been conducted with success for other bat-borne viruses such as lyssavirus (*i.e.*, *Lyssavirus hamburg*, EBLV-1) [59].

Previous studies have demonstrated that bat vesiculoviruses could be detected in different organs of infected individuals. For example, the American bat vesiculovirus has been identified in heart and lung homogenates, as well as in viscera pools of big brown bats (*Eptesicus fuscus*) [26]. The Chinese bat vesiculoviruses Jinghong bat virus and Benxi bat virus were detected in intestine and/or lung [27]. Similarly, the vesiculoviruses Yinshui bat virus, Taiyi bat virus and Qiongzhong bat virus, all identified in Chinese *Rhinolophus* bat species, have been also detected in different organs (including brain, heart, liver, spleen, lung, kidney and intestine) [28].

In our study, we did not have access to any cadavers or organs from positive specimens. However, MBV was detected in two different types of biological samples, with oral swabs for *Miniopterus schreibersii*, and with blood pellets for both *Rhinolophus euryale* and *Rhinolophus ferrumequinum*. Unfortunately, only one type of samples was available for each bat species, which did not allow us to investigate the presence of MBV in the oral swab of the two *Rhinolophus* species in Algeria and Morocco, or conversely to test for MBV in the blood of *Miniopterus* species in Spain. Although limited, these data evidence that MBV can be excreted via saliva, at least in *Miniopterus* species, which could play a role in virus

transmission. Unfortunately, oral swabs were placed directly into TRIzol after collection, making it impossible to test this hypothesis using viral isolation tests. Only a few brains from individuals collected in Algeria have been tested, all negative, but their small number does not allow us to conclude that MBV is not neurotropic.

The additional detection of MBV in blood pellets for *Rhinolophus euryale* and *Rhinolophus ferrumequinum* strongly suggest a viremia stage in both species. Most of vesiculovirus species was demonstrated to be transmitted by arthropods, mainly insects, which feed on the blood of infected vertebrates [60]. Some studies have also demonstrated that these ectoparasites can host numerous viruses, including rhabdoviruses. For example, short rhabdoviral sequences were detected in different ectoparasites (*Nycteribia kolenatii*, *Nycteribia schmidli* and *Penicillidia conspicua*) collected on Spanish bats [43]. In Uganda, two novel ledanteviruses, Bundibugyo and Kanyawara viruses, were identified in nycteribiid bat flies infesting pteropodid bats [33,36]. Kanyawara virus has been also detected directly in one of these infested bats. Thus, bat flies or other arthropods may serve as ectoparasitic reservoirs of "bat-associated" viruses, which can play a direct role in the virus circulation in bats (similar to that observed for most vesiculoviruses and other mammals), or only transiently or sporadically infect bats (as appears to be the case with Kanyawara virus) [36]. In our case, this could explain why MBV is found in different bat species sharing the same habitats and may be in close contact. However, as most bat flies remain host-specific to their host, further research is needed to assess this hypothesis. So far, the presence of the MBV in bat flies (*Nycteribiidae*) or ticks has not been investigated. In addition, attempts to isolate the virus from the remaining blood pellets, when still available, have not been successful. Such data will enable to test the hypothesis that MBV may be an arbovirus, and help to decipher the epidemiological cycle of MBV.

Several factors may explain the failure of virus isolation from blood. Sample quality may be one of them, particularly in the case of blood tube bottoms from a retrospective collection, which may have undergone different freeze/thaw cycles. Moreover, the quantity available was very limited. New prospective samples would be required to ensure optimal conditions for success. In addition, alternative strategies need to be pursued, including the use of more appropriate substrates such as cell lines from the species concerned (*i.e. Miniopterus schreibersii*, *Rhinolophus euryale* and *Rhinolophus ferrumequinum*). In fact, MBV cell receptors are not known and could be more or less specific to different bat species. Similarly, arthropod cell lines could be used for virus isolation, although the potential arthropod vectors remain unknown.

The presence of MBV in the different positive individuals was not associated with any clinical manifestation or symptoms, these animals having been captured in flight using nets, then released after sampling. Another recent study also described the presence of vesiculovirus infection in Chinese bats without any clinical signs at the time of capture [28]. More generally, various studies have demonstrated virus infection in asymptomatic bats, underlying the fact that these animals are playing a major role as potential virus reservoirs, with zoonotic potential for some of them. [4,6,61]. The current hypothesis is that a balanced immune response would enable them to maintain homeostasis during infection, limiting viral replication while avoiding the impact of excessive inflammation [62,63]. Deciphering these mechanisms, using adapted *in vitro* models, will enable us to assess and avoid the potential zoonotic risk of these animals, while paving the way for the development of therapeutics for infectious and inflammatory diseases.

## 5. Conclusion

Our study demonstrated that the prevalence of rhabdoviruses, detectable by our pan rhabdo nRT-qPCR, remained low within the bat samples tested. Despite this low prevalence, we were able to demonstrate the presence of a new species among the genus *Vesiculovirus*, which was associated to three bat species with *Miniopterus schreibersii*, *Rhinolophus euryale* and *Rhinolophus ferrumequinum*. These data confirm that a specific group of vesiculoviruses circulates throughout the Mediterranean region in insectivorous bats, without being apparently associated with disease symptoms in these animals. According to the virus genus considered (*Vesiculovirus*) and to the nature of the samples found positive (in particular blood pellets), we can speculate that MBV may be an arbovirus, like most other vesiculoviruses. However, further studies, particularly on arthropods, are needed to confirm this hypothesis.

MBV appears to be able to infect different bat species, unlike the other bat vesiculoviruses identified to date. So far, its ability to infect other mammals including humans, remains to be determined. However, the presence of this virus has been observed in oral swabs, suggesting that it could be excreted and transmitted via this mode, in addition to its circulating in the blood. These two factors, combined with the hypothesis that this virus can also be transmitted by arthropods, may increase the possibility of transmission to other animal species.

This potential zoonotic risk therefore requires more in-depth investigations, as do the mechanisms of infection and circulation in bat colonies. To achieve this goal, complementary analysis tools are in development, such as specific serological approaches, dedicated cellular models and reverse genetics approaches. These results also underline the role of bats in rhabdovirus diffusion and highlight the importance of performing active surveillance of this animal reservoir, a source of viral genetic diversity and potential rhabdovirus emergences.

## Supporting information

**S1 Fig. Multiple alignment of 103 nucleotide coding sequences of the polymerase of rhabdoviruses belonging to 14 different genera within the family *Rhabdoviridae*, with the position and the sequences of the four primers F1_Rhabdovirus, R1_Rhabdovirus, F2_Rhabdovirus-M and R2_Rhabdovirus selected for the pan-rhabdo RT-nqPCR.** The first PCR round is based on the primers F1_Rhabdovirus (forward, 5'-ATWGGNYTNAARS SIAARGA-3') and R1_Rhabdovirus (reverse, 5'-RYYTGRTTRTCNCCYTGIGC-3'), whereas the nested PCR round is based on primer F2_Rhabdovirus-M (forward, 5'-GAYTAYGANAARTGGAAYAAYYAYCA-3') and R2_Rhabdovirus (reverse, 5'-TGYCKNARNCCYTCYARNCCICC-3'). The nucleotide code applied is: R = A/G, Y = C/T, M = A/C, K = G/T, S = G/C, W = A/T, H = A/T/C, B = G/T/C, V = G/A/C, D = G/A/T, N = A/T/C/G and I (Hypoxanthine). The oligonucleotide sequence of each primer is indicated in bold, together with its name and an arrow indicating the sense direction. Sequence identity of the primers based on the multiple alignment is highlighted in black. The dot lines between the different blocks of sequence represent omitted regions. The position of the primers is indicated according to the nucleotide sequence of the polymerase gene of Drosophila obscura sigmavirus (DObSV) (GenBank accession number NC_022580). Virus acronyms and genera are indicated in the left of the figure. The multiple alignment was performed with ClustalW, version 2.0.
(DOCX)

**S2 Fig. Workflow analysis for the interpretation of the results obtained with the pan-rhabdo RT-nqPCR.** After RNA extraction of the samples, a first cDNA step is performed, followed by a 1st conventional PCR. A nested qPCR is systematically performed, and results are interpreted according to positive and negative samples, based on the shape of the melting curve and the value of the Tm. Positive samples are confirmed with an additional 2nd round of conventional PCR (from the 1st round conventional PCR). Amplicons are analyzed after migration on gel electrophoresis, and positive samples are Sanger sequenced. Validation of positive results is obtained after BLASTn and/or BLASTx analysis. Alternatively, amplicons from 2nd round of qPCR can also be directly Sanger sequenced and analyzed by BLAST. Templates (cDNA) from tissue samples need to be diluted 1:10 in nuclease-free water before the 1st round of conventional PCR. This dilution step is not required for liquid samples (*e.g.* blood or oral swab).
(DOCX)

**S3 Fig. Identification of the different deducted amino acid mutations between the 16 isolates of Mediterranean bat virus (MBV), according to the geographical location of the caves.** Dot lines indicate the conserved regions not shown in the figure. A total of 28 amino acid mutations were identified and represented. The multiple alignment was performed with ClustalW 2.0.
(DOCX)

**S1 Text. Supplementary materials and methods information.**
(DOCX)

**S1 Table. Details of the bat samples included in this study, according to the species and to the type of samples (brain, oral swab and blood).**
(DOCX)

**S2 Table. Description of the animal rhabdoviruses selected to design the primers of the pan-rhabdo RT-nqPCR based on the polymerase gene sequence.**
(DOCX)

**S3 Table. Description of the primers tested during the validation step of the different pan-rhabdovirus PCR systems.**
(DOCX)

**S4 Table. Description of the samples tested (rhabdovirus isolates and field samples) and the results obtained after the comparative evaluation of the different primers and qPCR systems.**
(DOCX)

**S5 Table. Results of pan-rhabdo RT-nqPCR detection with a panel of representative members of different genera of the family *Rhabdoviridae*.**
(DOCX)

**S6 Table. Specificity analysis of the pan-rhabdo RT-nqPCR on a panel of representative viruses other than rhabdoviruses and on a panel of non-infected samples.**
(DOCX)

**S7 Table. Specific primers designed to complete the full-length genome sequences of the bat Mediterranean vesiculovirus (MBV) isolates.**
(DOCX)

**S8 Table. List of vesiculoviruses and sequence identification selected for phylogenetic analysis of Mediterranean bat vesiculovirus isolates.** The lyssavirus RABV was selected as an outlier.
(DOCX)

**S9 Table. Details and NGS results of the 23 bat samples positive for rhabdovirus detection by the pan-rhabdo RT-nqPCR.**
(DOCX)

**S10 Table. Pairwise nucleotide comparison of the 16 bat rhabdovirus genomes expressed in nucleotide identity (%), after ORF concatenation for each of them.**
(DOCX)

**S11 Table. Amino acid identities (%) of the P, G and L proteins between the 16 isolates of Mediterranean bat virus (MBV).** Identities were calculated as pairwise deletion using MEGA7.0.
(DOCX)

**S12 Table. Amino acid identities (%) of the 5 proteins (N, P, M, G and L) of Mediterranean bat virus isolate 2012096 compared to other species among the genus *Vesiculovirus*.** Identities were calculated as pairwise deletion using MEGA7.0.
(DOCX)

## Acknowledgments

We wish to thank the Natural Park of Sant Llorenç del Munt i l'Obac (Barcelona, Spain), the National Park of Chréa (Algeria), the National Park of Béjaia (Algeria) and Drs. Mehdi Elharrak, Bachir Harif, Sehhar El Ayachi, Elbia Abdelatif and Ahmim Mourad for their precious contribution during field work. We would also like to thank all the other people who took part in the sample collection work in the field, without whom this study would not have been possible. We are also grateful to the Laboratory for Urgent Response to Biological Threats (CIBU) at Institut Pasteur, and in particular Christophe Batéjat, for providing us with different materials for the validation of the pan-rhabdo RT-nqPCR. Lastly, we thank Gaston for his excellent technical experience in bat design.

## Author contributions

**Conceptualization:** Dong-Sheng Luo, Laurent Dacheux.

**Data curation:** Dong-Sheng Luo, Laurent Dacheux.

**Formal analysis:** Dong-Sheng Luo, Laurent Dacheux.

**Funding acquisition:** Natalia Martinkova, Zheng-Li Shi, Hervé Bourhy, Laurent Dacheux.

**Investigation:** Dong-Sheng Luo, Markéta Harazim, Simon Bonas, Jordi Serra-Cobo, Laurent Dacheux.

**Methodology:** Dong-Sheng Luo, Laurent Dacheux.

**Project administration:** Dong-Sheng Luo, Laurent Dacheux.

**Resources:** Dong-Sheng Luo, Markéta Harazim, Natalia Martinkova, Aude Lalis, Emmanuel Nakouné, Edgard Valéry Adjogoua, Mory Douno, Blaise Kadjo, Marc López-Roig, Jiri Pikula, Jordi Serra-Cobo.

**Software:** Corinne Maufrais.

**Supervision:** Laurent Dacheux.

**Validation:** Dong-Sheng Luo, Laurent Dacheux.

**Visualization:** Dong-Sheng Luo, Laurent Dacheux.

**Writing – original draft:** Dong-Sheng Luo, Laurent Dacheux.

**Writing – review & editing:** Dong-Sheng Luo, Zheng-Li Shi, Hervé Bourhy, Jordi Serra-Cobo, Laurent Dacheux.

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
