## [Decision Letter · Decision Letter 0]

Feb 09 2025

PNTD-D-24-01255Large circulation of a novel vesiculovirus in bats in the Mediterranean regionPLOS Neglected Tropical Diseases Dear Dr. Dacheux, Thank you for submitting your manuscript to PLOS Neglected Tropical Diseases. After careful consideration, we feel that it has merit but does not fully meet PLOS Neglected Tropical Diseases's publication criteria as it currently stands. Therefore, we invite you to submit a revised version of the manuscript that addresses the points raised during the review process. Please submit your revised manuscript within 30 days Feb 09 2025 11:59PM. If you will need more time than this to complete your revisions, please reply to this message or contact the journal office at plosntds@plos.org. Please include the following items when submitting your revised manuscript: * A rebuttal letter that responds to each point raised by the editor and reviewer(s). You should upload this letter as a separate file labeled 'Response to Reviewers '. This file does not need to include responses to any formatting updates and technical items listed in the 'Journal Requirements' section below. * A marked-up copy of your manuscript that highlights changes made to the original version. You should upload this as a separate file labeled 'Revised Manuscript with Track Changes '. * An unmarked version of your revised paper without tracked changes. You should upload this as a separate file labeled 'Manuscript '. If you would like to make changes to your financial disclosure, competing interests statement, or data availability statement, please make these updates within the submission form at the time of resubmission. Guidelines for resubmitting your figure files are available below the reviewer comments at the end of this letter. We look forward to receiving your revised manuscript. Kind regards, Tony Schountz, PhDAcademic EditorPLOS Neglected Tropical Diseases Mabel CarabaliSection EditorPLOS Neglected Tropical Diseases

Shaden Kamhawi

co-Editor-in-Chief

Paul Brindley

co-Editor-in-Chief

**Additional Editor Comments:** Both reviewers have identified minor issues with the manuscript. Please address each comment in the revised manuscript. **Journal Requirements:**

At this stage, the following Authors/Authors require contributions: Dong-Sheng Luo, Markéta Harazim, Corinne Maufrais, Simon Bonas, Natalia Martinkova, Aude Lalis, Emmanuel Nakouné, Edgard Valéry Adjogoua, Mory Douno, Blaise Kadjo, Marc López-Roig, Jiri Pikula, Zheng-Li Shi, Hervé Bourhy, Jordi Serra-Cobo, and Laurent Dacheux. Please ensure that the full contributions of each author are acknowledged in the "Add/Edit/Remove Authors" section of our submission form.

2) We ask that a manuscript source file is provided at Revision. Please upload your manuscript file as a .doc, .docx, .rtf or .tex. If you are providing a .tex file, please upload it under the item type LaTeX Source File and leave your .pdf version as the item type Manuscript.

4) You stated "Virus isolation was attempted in newborn suckling BALB/c mice (3 days old) (Charles River laboratories). For each positive sample, 20 to 50 µL of lysed whole blood were diluted into 100 µL sterile PBS, gently mixed and centrifuged at 5000 g for 10 min at 4℃. The supernatant (5 to 10 µL) was inoculated intracerebrally into 4-5 newborn mice per sample." Please insert an Ethics Statement. It must include:

i) The full name(s) of the Institutional Review Board(s) or Ethics Committee(s)

ii) The approval number(s), or a statement that approval was granted by the named board(s).

5) Please upload all main figures as separate Figure files in .tif or .eps format. For more information about how to convert and format your figure files please see our guidelines: 

6) We have noticed there is a reference to Table S13 on page 9. However, there is no corresponding file uploaded to the submission. Please incorporate it in the Supporting Information file or if it is no longer to be included as part of the submission, please remove all reference to it within the text. 

7) We have noticed that you have uploaded Supporting Information files, but you have not included a list of legends. Please add a full list of legends for your Supporting Information files after the references list.

8) Some material included in your submission may be copyrighted. According to PLOS copyright policy, authors who use figures or other material (e.g., graphics, clipart, maps) from another author or copyright holder must demonstrate or obtain permission to publish this material under the Creative Commons Attribution 4.0 International (CC BY 4.0) License used by PLOS journals. Please closely review the details of PLOSu2019s copyright requirements here: PLOS Licenses and Copyright. If you need to request permissions from a copyright holder, you may use PLOS's Copyright Content Permission form.

Potential Copyright Issues:

i) Figure 3. Please confirm whether you drew the images / clip-art within the figure panels by hand. If you did not draw the images, please provide (a) a link to the source of the images or icons and their license / terms of use; or (b) written permission from the copyright holder to publish the images or icons under our CC BY 4.0 license. Alternatively, you may replace the images with open source alternatives. See these open source resources you may use to replace images / clip-art:

ii) Figure 1. Please (a) provide a direct link to the base layer of the map (i.e., the country or region border shape) and ensure this is also included in the figure legend; and (b) provide a link to the terms of use / license information for the base layer image or shapefile. We cannot publish proprietary or copyrighted maps (e.g. Google Maps, Mapquest) and the terms of use for your map base layer must be compatible with our CC BY 4.0 license.

9) We note that your Data Availability Statement is currently as follows: "All relevant data are within the manuscript and its Supporting Information files.". Please confirm at this time whether or not your submission contains all raw data required to replicate the results of your study. Authors must share the “minimal data set” for their submission. PLOS defines the minimal data set to consist of the data required to replicate all study findings reported in the article, as well as related metadata and methods (https://journals.plos.org/plosone/s/data-availability#loc-minimal-data-set-definition).

10) Please amend your detailed Financial Disclosure statement. This is published with the article. It must therefore be completed in full sentences and contain the exact wording you wish to be published.

11) Please ensure that the funders and grant numbers match between the Financial Disclosure field and the Funding Information tab in your submission form. Note that the funders must be provided in the same order in both places as well. Currently, the order of the funders is different in both places.

Please indicate by return email the full and correct funding information for your study and confirm the order in which funding contributions should appear. Please be sure to indicate whether the funders played any role in the study design, data collection and analysis, decision to publish, or preparation of the manuscript.

**Reviewers' comments:** Reviewer's Responses to Questions

**Key Review Criteria Required for Acceptance?**

**Methods**

-Are the objectives of the study clearly articulated with a clear testable hypothesis stated?

-Is the study design appropriate to address the stated objectives?

-Is the population clearly described and appropriate for the hypothesis being tested?

-Is the sample size sufficient to ensure adequate power to address the hypothesis being tested?

-Were correct statistical analysis used to support conclusions?

-Are there concerns about ethical or regulatory requirements being met?

Reviewer #1: (No Response)

Reviewer #2: We do not feel that any new analyses are needed. However, there are a number of methodological and experimental details that could be clarified and condensed.

The molecular methods (e.g. cDNA synthesis) are admirably thorough, but a lot of the standard methodology can be moved to the supplement and authors can highlight where their protocol deviated from the manufacturer's instructions. We appreciated them sharing these details and they will no doubt be useful to some but for the sake of the flow of the paper and people who are not directly replicating their experiments, it would be better in the supplement.

152-154: subordinate clause is missing a verb.

252: should be “were” instead of “was”

251-253 is basically identical to 261-263.

Section 2.1 and 2.2: were all samples preserved and stored in the same way prior to their shipment to Institut Pasteur? This could seriously impact sample degradation.

Section 2.3-2.5: positive and negative controls for virus are included, but was there any confirmation of successful RNA extraction/cDNA synthesis? This is important to determine whether negatives are true negatives or lack of starting material.

3. Geographic Focus of Samples:

○ The manuscript focuses on samples from Europe and Africa, but the rationale for this geographic selection is not clear. Why were these particular samples chosen? Is there a known difference in rhabdovirus prevalence or diversity in these regions that makes them particularly relevant to the study? A more thorough explanation of the importance of these geographic regions in the context of viral emergence and zoonotic transmission would be helpful.

**Results**

-Does the analysis presented match the analysis plan?

-Are the results clearly and completely presented?

-Are the figures (Tables, Images) of sufficient quality for clarity?

Reviewer #1: (No Response)

Reviewer #2: They did a great job testing their assay and the results seem robust though we would like clarity on whether sample degradation might have impacted their detection results. Other comments below:

Line 300: It’s an interesting finding that all brains were negative, but they aren’t discussed after this point.

Despite the assay being pan-rhabdovirus there were no detections of lyssaviruses even though some samples were originally collected for the purpose of lyssavirus monitoring. Were bats with EBLV or other lyssaviruses intentionally excluded? If not, is the lack of finding of EBLV surprising?

Figure 3: the labels of the grey ones especially the dark grey get hard to see; can you pull the text in front? Also it’s a little hard to tell what the boostrap values refer to in the R. affinis and R. sinicus clades.

372-378 and figure S3: If in the figure the authors could denote where each sequence is from it would be easier for the reader. Also are the mutations listed here by geography referring to how many sites with mutations there were in these areas?

**Conclusions**

-Are the conclusions supported by the data presented?

-Are the limitations of analysis clearly described?

-Do the authors discuss how these data can be helpful to advance our understanding of the topic under study?

-Is public health relevance addressed?

Reviewer #1: (No Response)

Reviewer #2: In general the conclusions are supported. We think the authors could improve the manuscript by streamlining it and removing interesting but irrelevant information (e.g. shape of rhabdoviruses) and focusing on the main applicability of their studies to infectious disease and public health. In particular, we highlight a few areas and specific lines:

Missing Emphasis on Importance: The broader significance of the study, especially in relation to viral emergence, zoonotic transmission, and the role of bats in rhabdovirus ecology, is not adequately highlighted. In particular, the importance of establishing a pan-animal rhabdovirus detection system should be emphasized more clearly. This could help in monitoring viral emergence and preventing zoonotic spillover, especially given the increasing frequency of novel virus discovery in bat populations.

Importance of Bats: The role of bats as key hosts for rhabdoviruses is mentioned in the introduction but should be given greater emphasis throughout the manuscript. The statement in the third paragraph of the introduction line 109 (“All together, these data suggest bats are playing an important role in the diffusion and persistence of rhabdoviruses…”) is crucial and should be moved earlier to frame the entire study. The evolutionary and viral emergence importance of bats as reservoirs for rhabdoviruses warrants further discussion to underscore the relevance of the tool being developed.

Lack of Discussion on Viral Evolution: While the manuscript notes the diversity of the Rhabdoviridae family, there is insufficient discussion on how this diversity contributes to viral evolution, including potential for zoonotic spillover or adaptation. A deeper exploration of how understanding rhabdovirus evolution could help mitigate risks associated with viral emergence would strengthen the manuscript.

Line 479: “we did not have access to any cadavers or organs” but brains are mentioned in the results. Again, this is an interesting discussion point to include. What could it mean that all brains were negative, but virus was detected in the blood and saliva?

What is the connection between this vesiculovirus and potential spillover to other species? Is there a zoonotic threat?

Make the connection between this vesiculovirus and potential spillover clearer.

I don’t think the results of this study reflect a “large” circulation of MBV; the title should be edited. Maybe "regional" circulation; the word "large" is a little misleading since it can also imply high prevalence. or "broad geographic" circulation?

444-449: Interesting there doesn’t seem to be host specificity. How does this compare to other vesiculoviruses?

500-501: I don’t disagree that this is possible but generally bat flies are pretty host specific. This seems likely to happen occasionally but not regularly.

Curious that none of them are positive for a lyssavirus given they were part of a screening program in Europe; were lyssavirus positive samples excluded?

**Editorial and Data Presentation Modifications?**

Reviewer #1: Please revise the "saliva samples" to oral swab! In fact, when swabbing bats, not only will there be saliva in the sample, but also cells and potentially remnants from ingested food, in bats most likely insects.

L 69: Please reconsider the use of virus vs. virus species. The latter is only the taxonomical unit, but is not "isolated" or can cause disease.

L 282/283: sensibility should be sensitivity

L 487: Remove In addtion

Reviewer #2: There’s a lot of repetition within the text across different sections, and the writing could overall be more concise. There are also minor grammatical errors throughout the text. Some are highlighted below.

65-67: This is kind of vague and bland; can the authors connect it more directly to their study?

79-80: Lyssaviruses are very extensively investigated in bats; I’d add the caveat Rhabdoviridae outside of lyssaviruses.

Line 81-112: this can be more concise

124: Should be “circulates” because the subject of the sentence is singular.

152-154: subordinate clause is missing a verb.

282-283: Test sensitivity instead of sensibility?

313-315: Not sure what this means; what is interesting here?

319: check the grammar

323: typo

356: define ABV

400: higher than those found in other studies?

404: Plecotus auritus is misspelled.

487: A sentence is began “In addition” but never completed.

**Summary and General Comments**

Reviewer #1: The authors developed a novel nested realtime PCR, screened existing sample sets from bats from various countries and identified novel rhabdoviruses, surprisingly at a low prevalence. Using state-of--the-art technologies the authors further characterised the identified sequences as a novel virus.

I congratulate the authors to this very fine work. In all sections, i.e. introduction, materials and methods, results and discussions, the authors provide a great level of detail and scientific accuracy.

Reviewer #2: Note: This review was conducted by a PI and two PhD students (hence the inconsistent use of plural and singular first person). We met to discuss our reviews and comments from all three people are aggregated.

The manuscript investigates the development of a novel molecular detection tool designed to identify rhabdoviruses in animal populations, with a particular focus on bat species. Using a combined nested RT-qPCR technique, the authors aim to address the significant challenge of rhabdovirus diversity, which complicates the detection of these viruses. The study attempts to bridge a gap in molecular diagnostics for rhabdoviruses, which are important pathogens for public health and veterinary concern, particularly in the context of viral emergence and zoonotic transmission. This manuscript presents a novel diagnostic tool and straightforward results; this paper would be valuable to the broader scientific community.

1. Novelty and Innovation: The development of a broad-spectrum molecular tool for the detection of rhabdoviruses for use across animal species is a valuable contribution to the field, particularly given the high genetic diversity within the Rhabdoviridae family. The use of nested RT-qPCR represents a solid technical approach for addressing this challenge that could be applied in laboratory settings with minimal equipment requirements.

2. Relevance to Public Health: The manuscript addresses an important gap in diagnostics for rhabdoviruses, which pose significant threats to both human and animal health. The potential for zoonotic transmission is a critical concern, and the development of a reliable screening tool could significantly improve monitoring efforts.

In general, we are excited about the diagnostic tool. The authors did an admirable job of vetting it to confirm it is truly pan-rhabdovirus and show interesting results of a new vesiculovirus circulating in Mediterranean bats. We think it will make a valuable contribution to the literature but recommend greater clarity about the sample preservation and lack of discovery of lyssaviruses or rhabdoviruses in the brains. We also recommend the authors streamline the manuscript to focus on the applicability of their diagnostic test and its potential role in understanding potential viruses of concern for bat or human health. (Or not! I don't know whether vesiculoviruses are likely to be zoonotic.)

PLOS authors have the option to publish the peer review history of their article (what does this mean? ). If published, this will include your full peer review and any attached files.

**Do you want your identity to be public for this peer review?** For information about this choice, including consent withdrawal, please see our Privacy Policy .

Reviewer #1: No

Reviewer #2: No

---

## [Editor Report · Decision Letter 1]

Dear Dr. Dacheux,

We are pleased to inform you that your manuscript 'Broad geographical circulation of a novel vesiculovirus in bats in the Mediterranean region' has been provisionally accepted for publication in PLOS Neglected Tropical Diseases.

Best regards,

Tony Schountz, PhD

Academic Editor

Shaden Kamhawi

co-Editor-in-Chief

Paul Brindley

co-Editor-in-Chief

---

## [Editor Report · Acceptance letter]

Dear Dr. Dacheux,

We are delighted to inform you that your manuscript, "Broad geographical circulation of a novel vesiculovirus in bats in the Mediterranean region," has been formally accepted for publication in PLOS Neglected Tropical Diseases.

Best regards,

Shaden Kamhawi

co-Editor-in-Chief

Paul Brindley

co-Editor-in-Chief
